# Peri-abortion contraceptive counseling: A systematic review of randomized controlled trials

Patricia Gonzales-Huaman[1], Jose Ernesto Fernandez-Chinguel[2], Alvaro Taype-Rondan [ID][3]*

1 Universidad Peruana de Ciencias Aplicadas, Escuela de Medicina, Lima, Peru, 2 Universidad de San Martín de Porres, Chiclayo, Peru, 3 Universidad San Ignacio de Loyola, Unidad de Investigación para la Generación y Síntesis de Evidencias en Salud, Lima, Peru

* alvaro.taype.r@gmail.com

**Data Availability Statement:** All relevant data are within the paper and its Supporting Information files. Data can also be accessed at https://figshare.com/s/a62bcb6af46d3e692327.

## Abstract

### Objective

To assess the effects of peri-abortion contraceptive counseling interventions.

### Methods

We performed a systematic review of randomized controlled trials (RCTs) that compared the effect of different types of peri-abortion contraceptive counseling interventions and were published as original papers in scientific journals. The literature search was performed in June 2021 in PubMed, Central Cochrane Library (CENTRAL), Scopus, and Google Scholar; without restrictions in language or publication date. Two independent authors identified studies that met the inclusion and exclusion criteria and extracted the data. The risk of bias was assessed using the Cochrane tool, and evidence certainty was assessed using the Grading of Recommendations Assessment, Development, and Evaluation (GRADE) methodology. Whenever possible, meta-analyses were performed. The protocol was registered at PROSPERO (CRD42020187354).

### Results

Eleven RCTs were eligible for inclusion (published from 2004 to 2017), from which nine compared enhanced versus standard counseling. Pooled estimates showed that, compared to standard counseling, enhanced counseling was associated with a higher incidence of effective contraceptive use (>3 months) (relative risk [RR], 1.12; 95% confidence interval [CI], 1.09–1.16), although no significant difference was found in the incidence of long-acting reversible contraceptive use (RR, 1.25; 95% CI, 0.68–2.29), contraceptive uptake (RR, 1.06; 95% CI, 0.98–1.15), and obstetric event occurrence (RR, 0.91; 95% CI, 0.57–1.47). Certainty of evidence was very low for all outcomes. In addition, two studies compared contraceptive counseling provided by physicians versus that provided by non-physicians, which did not show significant differences.

**Funding:** Self-financed.

**Competing interests:** The authors declare that they have no conflicts of interest regarding the subject of study.

## Conclusions

Enhanced contraceptive counseling may favor effective contraceptive use but may not affect the rate of obstetric event occurrence. Also, the studies did not find a difference in the effects of counseling interventions given by different providers. Since evidence certainty was very low, future well-designed RCTs are needed to make informed decisions.

## Registration

The study protocol was registered at PROSPERO (CRD42020187354).

## Introduction

Worldwide, from 2015 to 2019, approximately 121.0 million women had unintended pregnancy annually, of which 61% ended in abortion [1]. After a safe abortion procedure (either medical or surgical), fertility is not compromised, and women can start ovulation as early as eight days after abortion [2]. Thus, the risk of a further unintended pregnancy and abortion is not negligible [3].

Peri-abortion contraceptive counseling (either shortly before or shortly after abortion) could help prevent unintended pregnancies and other abortions [4]. However, these counseling interventions are provided using different strategies, structures, content, and healthcare providers [5] and seem to show heterogeneous results [6, 7].

Previous systematic reviews that evaluated peri-abortion contraceptive counseling [3, 6] were performed several years ago and did not assess the certainty of the evidence. Therefore, we performed a systematic review to summarize the available data from randomized controlled trials (RCTs) that evaluated the effects of peri-abortion contraceptive counseling interventions.

## Methods

### Design, protocol information, and patient involvement

We performed a systematic review, which was written according to the PRISMA 2020 statement. The study protocol was registered at PROSPERO (CRD42020187354), and there were no subsequent changes to the protocol. Raw data of the included studies can be accessed at https://figshare.com/s/a62bcb6af46d3e692327. Of note, patients and other relevant actors were not involved in this review

### Eligibility criteria

We performed a systematic review that assessed all RCTs that were published as original papers in scientific journals, which compared the effect of different types of peri-abortion (either shortly before or shortly after abortion) contraceptive counseling interventions (which could involve material and/or human resources). We defined standard counseling as the intervention regularly practiced in the study context and enhanced counseling as the new intensified strategy considered for the trial. No restrictions in language or publication date were applied.

### Information sources and search strategy

We performed a literature search of four sources: PubMed, Central Cochrane Library (CENTRAL), Scopus, and Google Scholar. Since Google Scholar sorts its results starting with those

that have the best match with the search terms, we consider that evaluating the first 100 results would include all relevant studies on the subject in this repository, a methodology that has been used in previous systematic reviews [8–11].

We searched the articles using the terms "counseling" (A), "abortion" (B), and "randomized controlled trial" (C), with the following syntax: A AND B AND C. The detailed search strategies for each search source are available in **S1 Table**.

## Selection process

The search was performed in two steps: 1) systematic search in four databases and 2) review of all references of the studies included in Step 1.

For step 1, we performed a literature search in June 2021, downloaded all results to an End-Note X8 document, and eliminated duplicated articles using this software. Subsequently, we assessed the titles and abstracts of each reference to identify potential studies for inclusion. Lastly, we assessed the full text of these potential studies to determine their eligibility.

For Step 2, we reviewed all references of studies that were included in Step 1 and collected new articles that met the inclusion criteria.

Both steps were performed independently by two authors. When disagreements were found, they were discussed by all authors and resolved by consensus.

## Data collection process

Two authors independently extracted the following information for the included studies into a Microsoft Excel sheet: author, year of publication, year of collection, title, country, population (inclusion and exclusion criteria), setting, peri-abortion contraceptive counseling given to the intervention and comparator groups (using the Template for Intervention Description and Replication (TIDieR) data extraction tool) [12], and results for all outcomes assessed in the studies. In case of disagreements, the full-text articles were reviewed again by all authors.

## Outcomes of interest

We evaluated the effect of contraceptive counseling interventions on contraceptive use (when an effective contraceptive method was used for more than three months), contraceptive uptake (when selected immediately after counseling), and obstetric event recurrence.

We considered effective contraceptive methods as any of the following: oral contraceptives, patch, ring, monthly injectable, quarterly injectable, condom, implant, vaginal ring, contraceptive patch, intrauterine device (IUD), intrauterine system (IUS), and sterilization or vasectomy. We considered long-acting reversible contraceptives (LARC) as any of the following: IUD, IUS/hormonal IUD, and implant. We considered obstetric events as either an unintended pregnancy or another induced abortion after counseling.

Evaluation time was considered as the moment in which the result was evaluated. This definition was applied for the outcomes of "use" and "uptake." Moreover, the follow-up time was considered as the maximum time that the participants were followed in each study. This definition was applied for the outcome of "occurrence of an obstetric event."

## Risk-of-bias assessment

To evaluate the risk of bias of included RCTs, we used the Cochrane risk-of-bias tool for randomized trials [13], which assesses the risk of bias in seven domains: random sequence generation, allocation concealment, blinding of participants and personnel, blinding of outcomes

assessment, incomplete outcome data, selective reporting, and other potential threats to validity.

## Synthesis methods

Whenever more than one study assessed a similar PICO (Population, Intervention, Control, and Outcome) question and the frequency of study events was greater than one in each group, we performed meta-analyses to summarize their results.

We assessed heterogeneity using an $I^2$ statistic, and heterogeneity was arbitrarily categorized using cutoff points agreed by the authors as not important ($I^2 < 40\%$), moderately significant ($I^2 = 40\%$–$75\%$), and considerable ($I^2 > 75\%$) [13]. We considered it appropriate to use random-effects models due to the overall heterogeneity of the interventions received by the participants in each study [14]. The data were processed using Review Manager 5.4 software.

Publication bias was not statistically assessed since the number of studies pooled for each meta-analysis was less than ten [14].

## Certainty assessment

To assess the certainty of the evidence for each outcome, we used the Grading of Recommendations Assessment, Development and Evaluation (GRADE) methodology [15, 16], which evaluates the study design, risk of bias, inconsistency, indirectness, imprecision, and publication bias.

## Results

### Study selection

We found 908 articles in the database search. After duplicates were removed, we screened 681 articles, from which 60 underwent full-text review. Then, 50 articles were excluded (detailed reasons for excluding these records are shown in **S2 Table**), and 10 studies were included [17–26]. Subsequently, we screened the 284 references of these 10 articles, from which one new study was included [27], for a total of 11 included studies (**Fig 1**).

### Study characteristics

Studies' characteristics are summarized in **Table 1** and presented at length in **S3 Table**. Moreover, the characteristics of the interventions in each study were identified in detail with the help of the TIDieR tool (**S5–S15 Tables**).

The 11 included studies were published between 2004 and 2017 but performed between 1999 and 2016. All were published in full-text in english. Three studies were performed in the USA [19, 20, 26], while the other eight were conducted in different countries. Regarding the continents, of 11 studies, four were performed in North America [19, 20, 23, 26], three in Europe [17, 22, 24], two in Asia [25, 27], one in East Africa [21], and one in South America [18].

Of the 11 included studies, seven showed the age means, ranging from 22.8 to 26.8 years. Furthermore, seven studies reported the frequency of previous abortion in their populations, ranging from 27.6% [18] to 50.5% [19]. The sample size ranged from 43 to 2336, and the follow-up period ranged from immediately to 16 weeks.

The included studies assessed two comparisons: nine studies compared enhanced contraceptive counseling and standard contraceptive counseling, provided to women before or after an abortion, while the other two studies compared abortion provision and contraceptive counseling, provided by physicians versus non-physician (nurses in one study and midwives

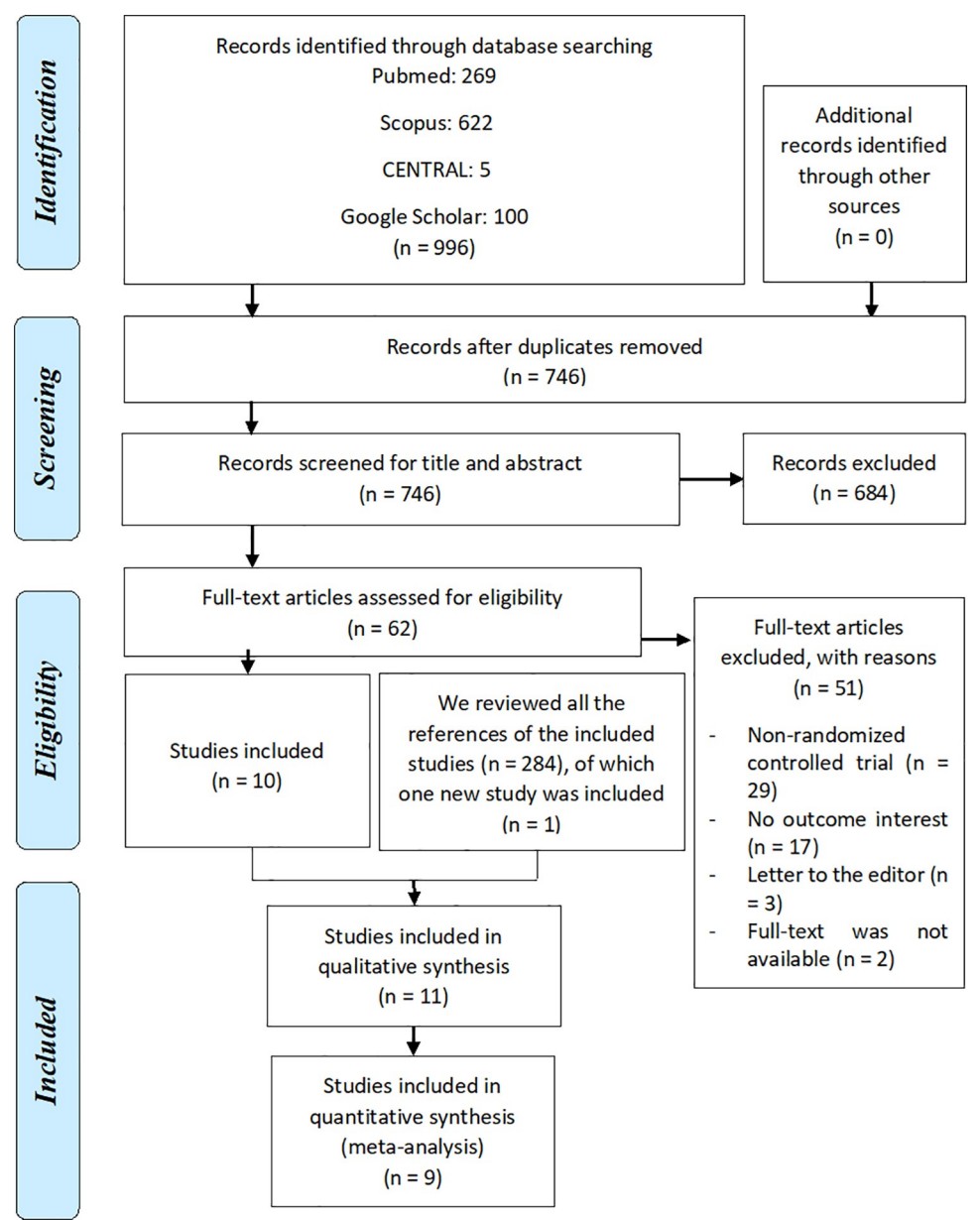

**Fig 1. Flow diagram (study selection).**

in the other study) (**Table 1**). Furthermore, of 11 studies, six reported that the intervention was adjusted to the needs of each patient [17, 18, 20, 22, 25, 26], three [23, 24, 27] did not specify it, and two were not [19, 21] (more details in **S5–S15** **Tables**).

## Risk of bias in studies

Among the 11 included studies, only one had a low risk of bias in all Cochrane Tool items (Whitaker 2016) [26], while nine adequately generated the randomization sequence, seven appropriately concealed allocation, three blinded participants and personnel, two blinded outcome evaluators, seven did not report a significant number of missing outcome data, and four had a protocol available where no selective outcome reporting was found (**Fig 2**).

**Table 1. Characteristics of included studies.**

| Author, year (country) | Randomized unit | Participants | Number of patients allocated (I, C) | Control group and intervention group | Maximum follow-up | Funding |
|---|---|---|---|---|---|---|
| Studies that have compared enhanced versus standard peri-abortion contraceptive counseling | | | | | | |
| Bender, 2004 (Iceland) | Women | Women who requested first-trimester pregnancy termination. | 276 (148, 128) | C: Routine counseling, defined as post-abortion routine contraceptive counseling, given once by a nurse or a midwife without special training in contraceptive counseling (duration not specified) + leaflets by a social worker + physician evaluation for contraception prescription. | 6 months | Not specified |
| | | | | I: Routine counseling + pre-abortion personalized contraceptive counseling given once by a specially trained family planning nurse (duration not specified) | | |
| Schunmann, 2006 (United Kingdom) | Weeks | Women presenting at the abortion clinic without an obstetric indication for pregnancy termination. | 377 (199, 178) | C: Standard care, defined as a pre-abortion brief discussion about contraception at the outpatient clinic given once (provider and duration not specified) + post-abortion contraceptive discussion with a nurse given once (duration not specified) and contraception provision. | 4 months | Not specified |
| | | | | I: Standard care + pre-abortion or post-abortion expert advice given once by a physician (interview lasted 10–20 min) and enhanced provision of certain contraception methods (three-month pills, implants, or IUD/IUS). | | |
| Nobili, 2007 (Italy) | Women | Women who requested pregnancy termination. | 43 (22, 21) | C: Standard care, defined as post-abortion referral of women to a community health center. | 3 months | Not specified |
| | | | | I: Standard care + pre-abortion personalized contraceptive counseling (consisting of a patient-centered interview, information, and education; election of a contraceptive method; and understanding checking), given once by a psychologist and gynecologist for 30 min. | | |
| Zhu, 2009 (China) | Hospital | Women seeking any abortion. | 1147 (592, 555) | C: Post-abortion essential package, defined as the provision of information for women in groups (times, provider, and duration not specified) and referral to women to existing family planning services | 6 months | EU 6th Framework Programme |
| | | | | I: Post-abortion comprehensive package, defined as a group and individual education for women and male involvement, free provision of contraception, and referral to existing family planning services given once (provider and duration not specified) | | |

(*Continued*)

**Table 1.** (Continued)

| Author, year (country) | Randomized unit | Participants | Number of patients allocated (I, C) | Control group and intervention group | Maximum follow-up | Funding |
|---|---|---|---|---|---|---|
| Langston, 2010 (USA) | Women | Women seeking a first-trimester abortion. | 222 (114, 108) | C: Standard care, defined as contraceptive counseling given once by a physician (content and duration left to the physician discretion). | 3 months | A grant from an anonymous foundation. |
| | | | | I: Standard care + pre-abortion standardized structured counseling using visual and audio material (duration not specified) and contraception provision given once by the research coordinator. | | |
| Carneiro, 2011 (Brazil) | Women | Women who had undergone an abortion. | 246 (123, 123) | C: Standard care, defined as post-abortion group educational counseling provided by a nurse for 30–40 min provided once and follow-up interview to verify use with gynecologist once. | 6 months | Not specified |
| | | | | I: Standard care + Post-abortion individually personalized three-stage counseling (education and information, guided information, free provision of chosen contraceptive, and verification of their understanding of their use) provided only once for 30 min by two trained providers. | | |
| Smith, 2015 (Cambodia) | Healthcare provider | Women who sought induced abortion and had a mobile phone | 500 (249, 251) | C: Standard care, defined as post-abortion family planning counseling at the clinic given once (provider and duration not specified) + the offer of a follow-up appointment at the clinic and the provision of a hotline number operated at Marie Stopes International Cambodia. | 12 months | The Marie Stopes International Innovation Fund and The UK Medical Research Council |
| | | | | I: Standard care + mobile phone-based intervention consisting of six automated interactive voice messages at the time of their preference by a counselor for three months. | | |
| Davidson, 2015 (USA) | Weeks | Women presenting for a surgical abortion. | 192 (97, 95) | C: Standard care, defined as pre-abortion contraception counseling given once by clinic staff + stress management video given once (duration not specified). | None | Grant Society Family Planning Research Funding |
| | | | | I: Standard care + long-acting contraception informative video given once by clinic staff (duration not specified). | | |
| Whitaker, 2016 (USA) | Women | Women seeking an abortion. | 60 (29, 31) | C: Standard care, defined as returning to usual care and receiving only non-standardized counseling given once by a clinic physician (duration not specified). | 3 months | National Health Institute |
| | | | | I: Standard care + seven-step motivational interview given once, provided by a physician or social worker (duration not specified). | | |

Studies that have compared peri-abortion contraceptive counseling given by physicians versus that given by non-physicians

*(Continued)*

**Table 1.** (Continued)

| Author, year (country) | Randomized unit | Participants | Number of patients allocated (I, C) | Control group and intervention group | Maximum follow-up | Funding |
|---|---|---|---|---|---|---|
| Olavarrieta, 2015 (Mexico) | Women | Women looking for medical abortion | 1017 (514, 503) | C: Abortion and post-abortion contraceptive method counseling provided by a physician who had recently joined the clinic staff and had never provided medical abortion or had only previously managed medical abortion under supervision, given once (duration not specified). | 15 days | Department of Reproductive Health and Research, UNDP, UNFPA, UNICEF, WHO, and the World Bank. |
| | | | | I: Abortion and post-abortion contraceptive method counseling provided by a nurse with no prior abortion experience, given once (duration not specified). | | |
| Makenzius, 2017 (Kennya) | Healthcare provider | Women with signs of incomplete abortion. | 810 (409, 401) | C: Abortion and post-abortion contraceptive counseling was given once (duration not specified) and was provided by a physician with a mean of 8.8 years of professional practice and 8.4 years of clinical experience in post-abortion counseling. | 10 days | The Swedish Research Council on Health, Working Life and Welfare |
| | | | | I: Abortion and post-abortion contraceptive counseling was given once (duration not specified) and was provided by a midwife with a mean of 22.4 years of professional practice and 2.7 years of clinical experience in post-abortion counseling. | | |

I, intervention; C, control.

## Summary of the results

We found that nine studies compared standard care and enhanced contraceptive counseling [17–20, 22, 24–27], while the other two studies compared contraceptive counseling provided by a physician versus non-physician [21, 23].

Regarding the comparison of standard and enhanced contraceptive counseling, the definition of each outcome differed across studies, as detailed in S4 Table. Moreover, each study had a particular way to enhance their regular contraceptive counseling, such as the addition of pre-abortion counseling sessions [17], enhancement of contraception provision [24], personalized contraceptive counseling [22], use of audiovisual material [19, 20], several stage counseling [18], mobile phone interventions [25], and motivational interview [26], as detailed in S5–S15 Tables.

Pooled analysis showed that, compared to standard care, enhanced contraceptive counseling might increase the incidence of effective contraceptive method use (eight RCTs; relative risk [RR], 1.12; 95% CI, 1.09–1.16; I2 = 93%), may have little to no effect on the incidence of LARC use (three RCTs; RR, 1.25; 95% CI, 0.68–2.29; I2 = 68%), may have little to no effect on the incidence of effective contraceptive method uptake (five RCTs; RR, 1.06; 95% CI, 0.98–1.15; I2 = 84%), and may have little to no effect on the incidence of obstetric event occurrence (three RCTs; RR, 0.91; 95% CI, 0.57–1.47; I2 = 63%); however, the evidence is very uncertain

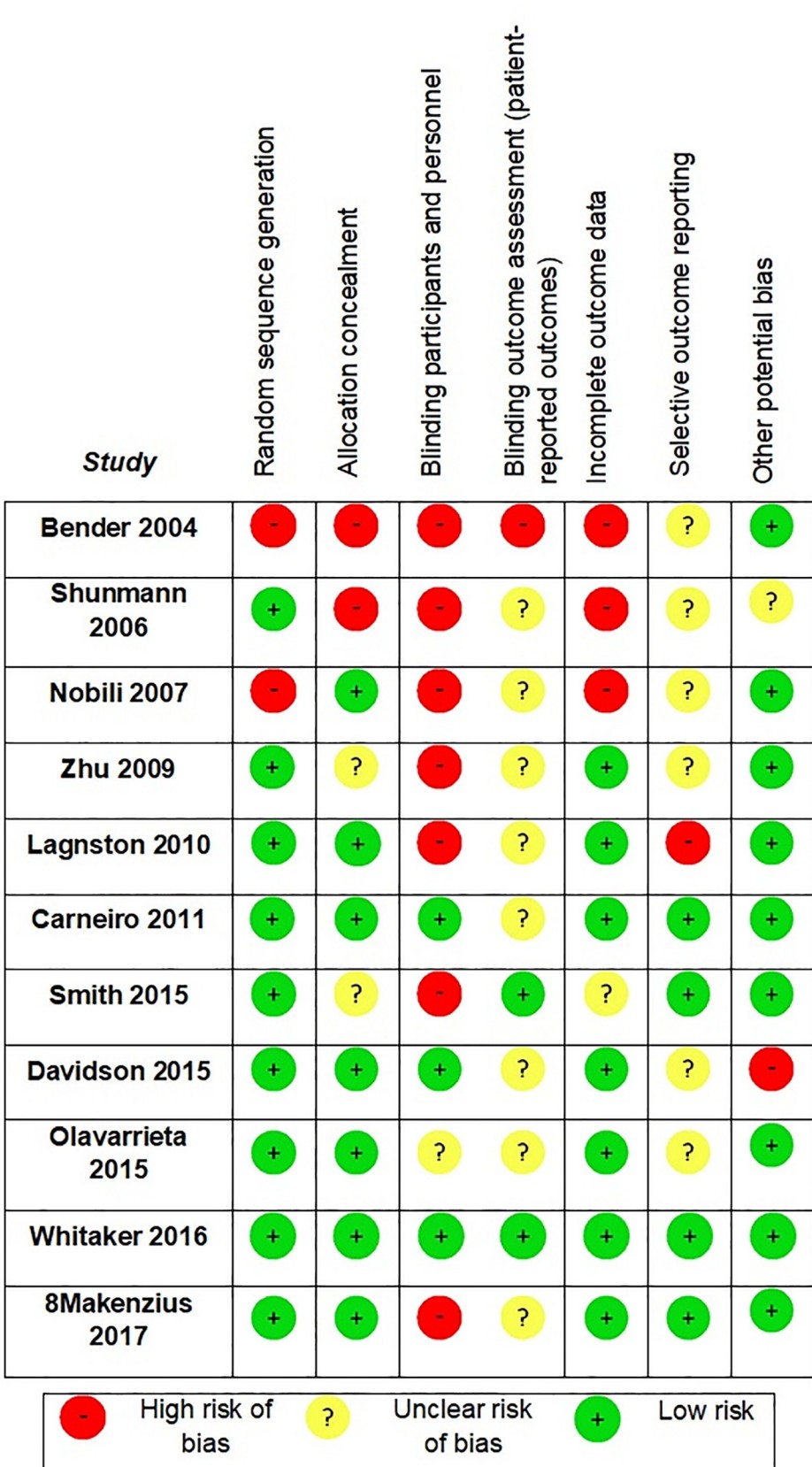

**Fig 2. Risk of bias included studies (cochrane tool).**

for these results. Meta-analyses are shown in **Fig 3**, and the Summary of Findings table is shown in **Table 2**.

Regarding the comparison of counseling provided by physicians and non-physicians, two RCTs were included: Olavarrieta 2015 (N = 884) and Makenzius 2017 (N = 803) [21, 23]. Due to the differences in the control group of the studies, we did not perform a meta-analysis.

Olavarrieta 2015 [23] compared counseling provided by physicians with counseling provided by nurses and did not find significant differences in the proportion of women prescribed contraceptives (nurse group: 99.1%, physician group: 98.7%), in the type of contraceptive prescribed, and in the proportion of women leaving the facility with at least one contraceptive method (96.7% vs 97.3%). However, the use of an intrauterine device was higher in the "physician group" than in the "nurse group" (31.3% vs 24.0%), while condom use was higher in the "nurse group" than in the "physician group" (19.2% vs 10.6%).

Makenzius 2017 [21] compared counseling provided by physicians and counseling provided by midwifes, and did not find significant differences in the proportion of women who received contraceptive counseling (midwife group: 98%, physician group: 98%), and in the accepted contraceptive method (74% vs 77%).

## Discussion

Two previous systematic reviews also assessed the effect of enhanced and standard peri-abortion contraceptive interventions [3, 6]. The systematic review of Ferreira [6] had its last search in 2007, which included three RCTs [17, 22, 24], and did not find any differences regarding contraceptive acceptance and use. The systematic review of Stewart [3] had its last search in 2014, included six RCTs [17, 18, 20, 22, 24, 27], and did not find differences in subsequent unplanned pregnancy rate, uptake of LARC, or continued use of selected contraceptive methods [3, 6]. Our review assessed all these outcomes and included a total of nine RCTs that compared standard and enhanced interventions, five of which were not in previous systematic reviews [3, 6].

### Contraceptive use

Regarding the contraceptive use outcome, the meta-analysis showed that the enhanced counseling group had a higher incidence of this outcome compared with the standard counseling group. However, study results were heterogeneous. Four studies had a high uptake incidence in their control group (Bender, 85.2%; Langston, 73.5%; Schunmann, 98.6%; Zhu, 89.4%), so the enhanced counseling group could not have a much higher incidence; therefore, it was not surprising that the intervention did not seem to be beneficial.

As enhanced interventions seemed to have a higher effect on contraceptive use, six studies primarily viewed personalized counseling as enhanced antisense counseling [17, 18, 20, 22, 25, 26]. Personalized counseling consisted mainly of providing information about different contraceptive methods [17, 20, 24, 25] for approximately 30 min [18, 22]. However, interventions were moderately heterogeneous across studies and varied in the provision of contraceptive methods, from studies providing free contraception [18, 27] to others providing a 3-month provision only for specific methods like pills, IUD, and implants [24].

The study that had more weight (45.4%) in the forest plot was Zhu's study, a study conducted in China, in which the intervention was a combination of individual and group counseling and included the women's significant others [27]; however, as most included studies in the meta-analysis [17, 20, 22, 24, 25, 27], the Zhu study has an overall unclear risk of bias due to failures in blinding of personal and participants.

## A. Use of effective contraception

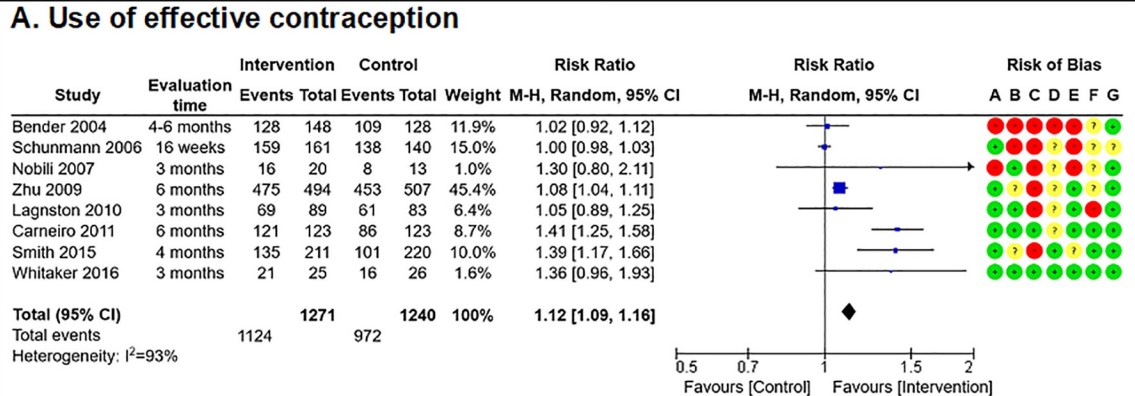

| Study | Evaluation time | Intervention Events | Total | Control Events | Total | Weight | Risk Ratio M-H, Random, 95% CI |
|---|---|---|---|---|---|---|---|
| Bender 2004 | 4-6 months | 128 | 148 | 109 | 128 | 11.9% | 1.02 [0.92, 1.12] |
| Schunmann 2006 | 16 weeks | 159 | 161 | 138 | 140 | 15.0% | 1.00 [0.98, 1.03] |
| Nobili 2007 | 3 months | 16 | 20 | 8 | 13 | 1.0% | 1.30 [0.80, 2.11] |
| Zhu 2009 | 6 months | 475 | 494 | 453 | 507 | 45.4% | 1.08 [1.04, 1.11] |
| Lagnston 2010 | 3 months | 69 | 89 | 61 | 83 | 6.4% | 1.05 [0.89, 1.25] |
| Carneiro 2011 | 6 months | 121 | 123 | 86 | 123 | 8.7% | 1.41 [1.25, 1.58] |
| Smith 2015 | 4 months | 135 | 211 | 101 | 220 | 10.0% | 1.39 [1.17, 1.66] |
| Whitaker 2016 | 3 months | 21 | 25 | 16 | 26 | 1.6% | 1.36 [0.96, 1.93] |
| **Total (95% CI)** | | | 1271 | | 1240 | 100% | **1.12 [1.09, 1.16]** |
| Total events | | 1124 | | 972 | | | |
| Heterogeneity: I²=93% | | | | | | | |

## B. Use of long-acting reversible contraceptives

| Study | Evaluation time | Intervention Events | Total | Control Events | Total | Weight | Risk Ratio M-H, Random, 95% CI |
|---|---|---|---|---|---|---|---|
| Bender 2004 | 4-6 months | 8 | 148 | 13 | 128 | 25.4% | 0.53 [0.23, 1.24] |
| Schunmann 2006 | 16 weeks | 60 | 157 | 35 | 136 | 43.0% | 1.48 [1.05, 2.10] |
| Whitaker 2016 | 3 months | 15 | 25 | 8 | 26 | 31.6% | 1.95 [1.01, 3.77] |
| **Total (95% CI)** | | | 330 | | 290 | 100% | **1.25 [0.68, 2.29]** |
| Total events | | 83 | | 56 | | | |
| Heterogeneity: I²=68% | | | | | | | |

## C. Uptake of an effective contraceptive method

| Study | Evaluation time | Intervention Events | Total | Control Events | Total | Weight | Risk Ratio M-H, Random, 95% CI |
|---|---|---|---|---|---|---|---|
| Schunman 2006 | Inmediatly | 271 | 281 | 203 | 250 | 23.0% | 1.19 [1.11, 1.27] |
| Lagnston 2010 | Inmediatly | 105 | 114 | 99 | 108 | 21.3% | 1.00 [0.93, 1.09] |
| Carneiro 2011 | Inmediatly | 123 | 123 | 118 | 123 | 25.4% | 1.04 [1.00, 1.08] |
| Davidson 2015 | Inmediatly | 92 | 96 | 91 | 95 | 23.4% | 1.00 [0.94, 1.06] |
| Whitaker 2016 | 4 weeks | 25 | 29 | 23 | 31 | 7.0% | 1.16 [0.90, 1.50] |
| **Total (95% CI)** | | | 643 | | 607 | 100% | **1.06 [0.98, 1.15]** |
| Total events | | 616 | | 534 | | | |
| Heterogeneity: I²=84% | | | | | | | |

## D. Occurrence of obstetric event (Pregnancies, unwanted pregnancies, and induced abortions)

| Study | Follow Up | Intervention Events | Total | Control Events | Total | Weight | Risk Ratio M-H, Random, 95% CI |
|---|---|---|---|---|---|---|---|
| Schunmann 2006 | 2 years | 40 | 316 | 27 | 297 | 36.5% | 1.39 [0.88, 2.21] |
| Zhu 2009 | 6 months | 56 | 592 | 82 | 634 | 44.3% | 0.73 [0.53, 1.01] |
| Smith 2015 | 1 year | 8 | 169 | 11 | 159 | 19.2% | 0.68 [0.28, 1.66] |
| **Total (95% CI)** | | | 1077 | | 1090 | 100% | **0.91 [0.57, 1.47]** |
| Total events | | 104 | | 120 | | | |
| Heterogeneity: I²=63% | | | | | | | |

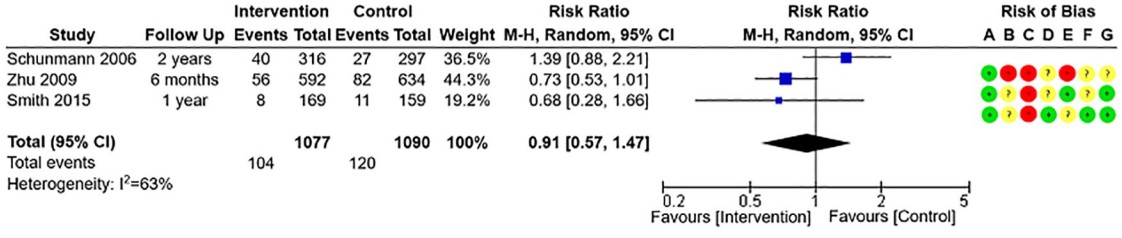

Risk of bias legend
(A) Random sequence generation (selection bias)
(B) Allocation concealment (selection bias)
(C) Blinding of participants and personnel (performance bias)
(D) Blinding of outcome assessment (detection bias)
(E) Incomplete outcome data (attrition bias)
(F) Selective reporting (reporting bias)
(G) Other bias

**Fig 3. Meta-analyses of studies that compared peri-abortion enhanced contraceptive counseling (intervention group) vs standard contraceptive counseling (control group).**

## Uptake

Regarding the uptake outcome, the meta-analysis did not show a significant difference among the standard and enhanced counseling groups. However, study results were heterogeneous. Similar to what was described for the use outcome, in this case, the studies in which enhanced counseling did not seem to have a benefit showed a higher use incidence in the control group (Carneiro, 95.9%; Davidson, 95.8%; Langston, 91.7%), so it was not surprising that the intervention did not seem to be beneficial in these studies. However, the other two studies, which seem to show a benefit in the enhanced group, had a lower uptake incidence in the control group (Schunmann, 81.20%; Whitaker, 74.19%). This may be because their "uptake" definition did not include some hormonal contraceptive types, such as implant and monthly injectable.

LARC use has been the main objective of contraception campaigns in the last years [28]. We found a slight trend in enhanced counseling compared to standard counseling, which is represented by two studies with a larger population; however, in the meta-analyses, we found that "contraceptive use" is similar in the enhanced counseling and standard counseling groups. Of the three meta-analyzed studies for LARC use, while two suggested a benefit of the enhanced intervention, Bender 2004 [17] did not. However, given this study's high risk of bias (including randomization problems causing the study groups to have different basal characteristics and lack of adjusted analysis for possible confounding factors), its results should be taken with caution. Currently, no clear consensus has been found that defines how long a person must be evaluated to guarantee the contraceptive method's effectiveness. We consider that the evaluation period was noticeably short in the studies as the outcome was a long-term method.

Enhanced counseling at the time of abortion seems to reduce the occurrence of obstetric events; however, it is necessary to consider that the certainty of the evidence for all outcomes was very low, so future studies are needed to confirm this result and elicit if this possible benefit may be due to patient-centered counseling, explanation of myths, or other components [6]

**Table 2. Summary of Findings (SoF) table.**

Studies design: Randomized controlled trials.
Population: Women who underwent an abortion.
Intervention: enhanced peri-abortion contraceptive counseling interventions.
Control: standard peri-abortion contraceptive counseling interventions.

| Outcomes | No. of participants and studies (I, intervention; C, control) | Standard care | Enhanced counseling | Relative effect (95% CI) | Risk difference (95% CI) | Certainty of the evidence (GRADE) |
|---|---|---|---|---|---|---|
| Use of effective contraception | I: 1124, C: 972 (8 RCTs) | 784 per 1000 | 878 per 1000 | RR 1.12 (1.09 to 1.16) | 94 more per 1000 (71 more to 125 more) | Very low [1,2,3] ⊕◯◯◯ |
| Use of long-acting reversible contraceptives | I: 83, C:56 (3 RCTs) | 193 per 1000 | 241 per 1000 | RR 1.25 (0.68 to 2.29) | 48 more per 1000 (62 fewer to 249 more) | Very low [1,2,4,5] ⊕◯◯◯ |
| Uptake of an effective contraceptive method | I: 616, C: 534 (5 RCTs) | 880 per 1000 | 933 per 1000 | RR 1.06 (0.98 to 1.15) | 53 more per 1000 (18 fewer to 132 more) | Very low [1,2,3] ⊕◯◯◯ |
| Occurrence of an obstetric event | I: 104, C: 120 (3 RCTs) | 112 per 1000 | 100 per 1000 | RR 0.91 (0.57 to 1.47) | 10 fewer per 1000 (47 fewer to 52 more) | Very low [1,2,5] ⊕◯◯◯ |

1. RCTs with a high risk of bias.

2. Intervention and control groups received different interventions in each study.

3. Heterogeneity $I^2 > 75\%$.

4. Wide confidence intervals.

5. Heterogeneity $I^2 > 40\%$.

and if a higher effect can be achieved with a group or individual interventions in which the significant other is included [27].

No previous systematic reviews studied the difference between counseling provided by a physician and that by a non-physician. We did not conduct a meta-analysis for this comparison since the studies were quite different. Olavarrieta and Makenzius showed that, compared to physicians, non-physicians tend to prescribe more contraceptives but have fewer users taking contraceptives home [21, 23] and prescribe fewer IUDs [23]. However, more studies are needed to make a reliable conclusion.

## Limitations and strengths

Some limitations must be considered when interpreting the results of this systematic review. Mainly, most RCTs did not show a minimally informative description of what counseling was provided to the control and intervention groups, which prevented them from fully understanding their results, which could be improved using TIDieR [12] and following the CONSORT checklist [29]. This is important since the effectiveness of counseling interventions may depend on several factors [30], including communication skills of the providers, considering key aspects of the patient's life to develop the intervention [31, 32], and considering the woman's family plan [6], need of information, and past experiences with contraception [7], in addition to their preferences that should be consistent with the contraceptive methods they use [33]. Likewise, the included studies did not report an evaluation of the counseling quality nor the user satisfaction.

Other important limitations were as follows: 1) few studies had a low risk of bias, and most failed in blinding. 2) Enhanced interventions were moderately heterogeneous across studies. It is expected to have certain heterogeneity since different contexts need different interventions. However, some counseling guidelines, such as those of the World Health Organization [20] and the United States Agency for International Development [34], can be used in future studies to establish certain components that are minimally assessed in contraceptive counseling interventions [34]. 3) Outcomes had different definitions across studies [7]. 4) Studies usually do not detail important information to understand their results, such as abortion restrictions and feasibility in their settings, although it seems that, of 11 studies, six were performed in countries where abortion is legal on request and three in a restrictive setting [7]. Future RCTs must consider these limitations.

Likewise, it is necessary to acknowledge that all included studies seem to have been performed considering sex or having a uterus and did not ask about gender identity, and we found no information regarding the effect of different types of counseling in specific groups, such as transgender man or nonbinary individuals, in which counseling may need to consider specific components [35].

However, to date, this is the most comprehensive systematic review that has assessed the effects of peri-abortion contraceptive counseling, summarizing information that may be useful to informed decision-making.

## Conclusion

We found that enhanced contraceptive counseling may increase the use of effective contraception but may not seem to affect the occurrence of obstetric events (pregnancies or abortions). Moreover, studies have not been able to find a difference in the effects of counseling interventions given by different providers. However, given that the certainty of the evidence was very low, future well-designed RCTs are needed to make an informed decision.

## Supporting information

**S1 Checklist. PRISMA 2020 checklist.**
(DOCX)

**S1 Table. Search terms.**
(DOCX)

**S2 Table. List of studies that were assessed in full-text and excluded.**
(DOCX)

**S3 Table. Characteristics of the studies, in extenso.**
(DOCX)

**S4 Table. Definition of the outcomes assessed in the meta-analyses in each study.**
(DOCX)

**S5 Table. Detail of the interventions received in Bender´s study.**
(DOCX)

**S6 Table. Detail of the interventions received in Schunmann´s study.**
(DOCX)

**S7 Table. Detail of the interventions received in Nobili´s study.**
(DOCX)

**S8 Table. Detail of the interventions received in Zhu's study.**
(DOCX)

**S9 Table. Detail of the interventions received in Lagnston´s study.**
(DOCX)

**S10 Table. Detail of the interventions received in Carneiro´s study.**
(DOCX)

**S11 Table. Detail of the interventions received in Davidson´s study.**
(DOCX)

**S12 Table. Detail of the interventions received in Smith´s study.**
(DOCX)

**S13 Table. Detail of the interventions received in Olavarrieta´s study.**
(DOCX)

**S14 Table. Detail of the interventions received in Whitaker´s study.**
(DOCX)

**S15 Table. Detail of the interventions received in Makenzius´s study.**
(DOCX)

**S1 Data.**
(XLSX)

## Author Contributions

**Conceptualization:** Patricia Gonzales-Huaman, Jose Ernesto Fernandez-Chinguel.

**Data curation:** Patricia Gonzales-Huaman, Jose Ernesto Fernandez-Chinguel, Alvaro Taype-Rondan.

**Formal analysis:** Patricia Gonzales-Huaman, Jose Ernesto Fernandez-Chinguel, Alvaro Taype-Rondan.

**Funding acquisition:** Patricia Gonzales-Huaman, Jose Ernesto Fernandez-Chinguel, Alvaro Taype-Rondan.

**Investigation:** Patricia Gonzales-Huaman, Jose Ernesto Fernandez-Chinguel, Alvaro Taype-Rondan.

**Methodology:** Patricia Gonzales-Huaman, Jose Ernesto Fernandez-Chinguel, Alvaro Taype-Rondan.

**Project administration:** Patricia Gonzales-Huaman, Jose Ernesto Fernandez-Chinguel, Alvaro Taype-Rondan.

**Resources:** Patricia Gonzales-Huaman, Jose Ernesto Fernandez-Chinguel, Alvaro Taype-Rondan.

**Software:** Patricia Gonzales-Huaman, Jose Ernesto Fernandez-Chinguel, Alvaro Taype-Rondan.

**Supervision:** Alvaro Taype-Rondan.

**Validation:** Patricia Gonzales-Huaman, Jose Ernesto Fernandez-Chinguel, Alvaro Taype-Rondan.

**Visualization:** Patricia Gonzales-Huaman, Jose Ernesto Fernandez-Chinguel, Alvaro Taype-Rondan.

**Writing – original draft:** Patricia Gonzales-Huaman, Jose Ernesto Fernandez-Chinguel, Alvaro Taype-Rondan.

**Writing – review & editing:** Patricia Gonzales-Huaman, Jose Ernesto Fernandez-Chinguel, Alvaro Taype-Rondan.

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
