## [Decision Letter · Decision Letter 0]

12 Oct 2021

PONE-D-21-29042Peri-abortion contraceptive counseling: a systematic review of randomized controlled trialsPLOS ONE

Estimado Taype-Rondan,

Thank you for submitting your manuscript to PLOS ONE. After careful consideration, we feel that it has merit but does not fully meet PLOS ONE’s publication criteria as it currently stands. Therefore, we invite you to submit a revised version of the manuscript that addresses the points raised during the review process.

We look forward to receiving your revised manuscript.

Saludos cordiales,

Dylan A Mordaunt, MB ChB, FRACP, FAIDH

Academic Editor

PLOS ONE

Journal Requirements:

"Self-financed" 

Additional Editor Comments:

Thank you for your submission. With regards to publication criteria (https://journals.plos.org/plosone/s/criteria-for-publication):

1) This study appears to present the results of original research.

2) Results reported do not appear to have been published elsewhere.

3) Experiments, statistics, and other analyses are performed to a high technical standard and are described in sufficient detail. There are some minor suggestions made by a number of reviewers before publication is accepted.

4) Conclusions are presented in an appropriate fashion and are supported by the data.

5) The article is presented in an intelligible fashion and is written in standard English.

6) The research meets all applicable standards for the ethics of experimentation and research integrity.

7) The article adheres to appropriate reporting guidelines and community standards for data availability.

Additional points:

- Thank you for submitting your systematic review in a structured format- there's a conflict between whether you've submitted in the 2009 or 2020 format of PRISMA.

- I would also suggest reviewing your manuscript with regards to the AMSTAR2 checklist (or an equivalent), to maximise the measured quality of this SR and consider including this as a supplementary file.

- It would be useful to clarify whether there are any differences between the PROSPERO protocol and the methods that were followed in the manuscript.

- The cut-offs used for heterogenity assessment and the choice to quantitatively synthesise are interesting- it would be useful to reference and give an explaination for why this was felt to be warranted as it could be a point of criticism.

Reviewers' comments:

Reviewer's Responses to Questions

**Comments to the Author**

1. Is the manuscript technically sound, and do the data support the conclusions?

Reviewer #1: Yes

Reviewer #2: Yes

Reviewer #3: Yes

Reviewer #4: Yes

Reviewer #5: Yes

Reviewer #6: Yes

2. Has the statistical analysis been performed appropriately and rigorously? 

Reviewer #1: Yes

Reviewer #2: No

Reviewer #3: Yes

Reviewer #4: I Don't Know

Reviewer #5: Yes

Reviewer #6: Yes

3. Have the authors made all data underlying the findings in their manuscript fully available?

Reviewer #1: Yes

Reviewer #2: Yes

Reviewer #3: Yes

Reviewer #4: Yes

Reviewer #5: Yes

Reviewer #6: Yes

4. Is the manuscript presented in an intelligible fashion and written in standard English?

Reviewer #1: Yes

Reviewer #2: No

Reviewer #3: Yes

Reviewer #4: No

Reviewer #5: Yes

Reviewer #6: Yes

5. Review Comments to the Author

Reviewer #1: Gonzales-Huaman and co-authors performed a systematic review about peri-abortion contraceptive counseling interventions.

The introduction builds a logical context for the article, the purpose of which is clear and well defined, appropriate physiopathological background is given, and key concepts are defined. I think the motivation for the present research is clearly expressed.

Data comprised in the tables is clear and consistent with the article’s content. The statistical methods used are appropriate.

The literature is analyzed and critically appraised. Ideas are acknowledged appropriately and accurately; there are no instances of plagiarism, and reference citations are complete and accurate.

Reviewer #2: 1. The article has not been published elsewhere.

2. The statistical analysis has been performed appropriately without performing a meta-analysis for counselling provided by physicians and midwives.

3. The article is not presented in an intelligible fashion or is written in standard English.

4. It is not necessary

5. The article adheres to appropriate reporting guidelines and community standards for data availability

Reviewer #3: This is a relevant systematic review which is intended to provide robust evidence regarding the effectiveness of peri-abortion contraceptive counseling in several settings towards reduction of the incidences of repeat abortions.

The manuscript has been well written with full considerations for the requirements for the condcut and reporting of sytematic reviews and meta analyisis.

There are however few gramatical errors that needs to be corrected and suggestiosn towards improving the quality of the report.

Please find below further comments on the manuscript.

ABSTRACT

Comment 1: The time frame for the studies included in the systematic review is conspicuously missing. This should be reflected not in the abstract.

Comment 2: Also include the databases searched by the authors ad whether there were language restrictions in the search.

Conclusion

Comment 3: The authors stated in the conclusion “There is no evidence that involvement of different professionals influences contraceptive use”. It is not clear how this was arrived at since only two studies out of 11 compared influence of the involvement of different professionals in counselling. Rather due to the paucity of RCT that compared the two approaches, there is limited evidence that involvement of different professionals influences contraceptive use.

Key messages

Comment 4: The key messages from this study has not been well highlighted in this section. The following statements should be deleted from this section during revision:

•We conducted a systematic review and meta-analysis of 11 RCTs to determine the effect of peri-abortion contraceptive counseling.

• We evaluated the effect of contraceptive counseling on peri-abortion contraceptive use to assess the contribution of this strategy to reducing future pregnancies.

Replace the above with information that highlights key messages like whether this was the first RCT on the subject matter, uniqueness of study, findings, conclusions, and recommendations:

Introduction

Page 4, lines 2-3: Globally, abortion is a complex situation for people with uterus (cis, trans and non-binary people). [2].

Comment 5: This statement may not be relevant to the context. Please delete

Page 4, Lines 14-17: Therefore, we performed a systematic review to summarize the available data on randomized controlled trials (RCTs) effects of peri-abortion contraceptive counseling intervention on contraceptive use and obstetric outcomes.

Comment 6: Paraphrase to: Therefore, we performed a systematic review to summarize the available data from randomized controlled trials (RCTs) that evaluated the effects of peri-abortion contraceptive counseling intervention on contraceptive use and obstetric outcomes

Methods

Page 4, line 24, Page 5 lines 1-2: and enhanced the new strategy considered for the trial) and were published as original papers in scientific journals

Comment 7: and enhanced counselling as the new strategy considered for the trial) and were published as original papers in scientific journals.

Information sources and search strategy

Page 5, lines 6-8: We performed a literature search of four sources: PubMed, Central Cochrane Library (CENTRAL), Scopus, and Google Scholar (for this, we only assessed the first 100 results).

Comment 8: It is not clear why only the first 100 results from google scholar was assessed. This should be elucidated upon.

Outcomes of interest

Comment 9: The authors have listed various definition of terms considered for this study but has not referenced these considerations for objectivity in assessment of these studies in relation to widely accepted definitions. This should be included in the revision.

Results

Page 11, lines 1-2: In addition, the proportion of women leaving the facility with at least one contraceptive method was similar in the “nurse group” than in the “physician group”

Comment 10: Change to: In addition, the proportion of women leaving the facility with at least one contraceptive method was similar in both groups

Discussion

This section has been well written and exhaustive

Page 11, line 18: which include six RCTs

Comment 11: which included six RCTs

Limitations

The authors have highlighted the main limitations to this study and interpretation of results.

Strength

However, to date, this is the most comprehensive systematic review that has assessed the effects of peri-abortion contraceptive counseling, summarizing information that may be useful to informed decision-making.

Comment 12: This message should be adapted as part of the key message from this study.

References

The journal names should be written as they are abbreviated in the list of index medicus for all the references quoted.

Reviewer #4: INTRODUCTION: There are some grammatical errors that need to be corrected. The aim of the study in the last paragraph of the introduction on page 4 is not clear.

RESULTS: Wrong use of tenses and few grammatical errors.

DISCUSSION: In page 12, paragraph 3, in the 1st sentence, what do the authors mean by ' the study had more weight'?

Reviewer #5: 1. Summary of Research

In this work, the authors sought to undertake a systematic review and meta-analysis of 11 RCTs to determine the effect of peri-abortion counseling. This review concluded that enhanced contraceptive counseling may favour effective contraceptive use but may not affect the rate of obstetric event occurrence, though the certainty of the evidence was low.

Below are some comments for the authors:

2. Specific Areas

Methods:

In the first paragraph under the under the sub-heading ‘outcome of interest’-page 6, authors should kindly note that all injectable contraceptives are hormonal. So the monthly and quarterly injectables are implied. Authors should kindly take note of this.

Again, Intrauterine system (IUS) is used interchangeable with hormonal IUD.

Language editing is recommended in the 2nd paragraph under the sub-heading ‘outcome of interest’. Specifically “we considered standard care or routine counseling to all interactions with patients preestablished by the hospital, institution or the research group”

Limitations and Strengths

Were there any limitations in the area of language bias and how was it addressed since the various studies were undertaken in different continents with heterogeneous cultural diversities.

3. Additional Comments

Inclusion of line numbers on manuscript could have been helpful for the review processes.

Reviewer #6: manuscript needs to be shortened. Authors should also engage a statistician. Consider using the forest plot. also can you include a table that shows the articles reviewed and results at a glance? thank you for great review

6. PLOS authors have the option to publish the peer review history of their article (what does this mean?). If published, this will include your full peer review and any attached files.

Reviewer #1: **Yes: **Daniela-Roxana Matasariu

Reviewer #2: No

Reviewer #3: **Yes: **DR GODWIN O. AKABA,MBBS,MSc,MPH,FWACS

Reviewer #4: No

Reviewer #5: **Yes: **Dr. Timothy K. Adjei

Reviewer #6: **Yes: **Emmanuel Ugwa

---

## [Author Response · Author response to Decision Letter 0]

13 Nov 2021

Here we answer to each of the reviewers commentaries:

Journal Requirements:

o We have made the changes according to the PlosOne templates.

o We did a reference check and we found that the reference list is complete and correct.

• 3. Thank you for stating the following financial disclosure: "Self-financed" 

o Ok, please tell us if any other detail is required.

Additional points:

• Thank you for submitting your systematic review in a structured format- there's a conflict between whether you've submitted in the 2009 or 2020 format of PRISMA.

o We follow the format of PRISMA 2020 format.

• I would also suggest reviewing your manuscript with regards to the AMSTAR2 checklist (or an equivalent), to maximise the measured quality of this SR and consider including this as a supplementary file.

o We agree. We attached as a supplementary file such AMSTAR 2 evaluation.

• It would be useful to clarify whether there are any differences between the PROSPERO protocol and the methods that were followed in the manuscript.

o We agree. We added the following as the first paragraph of the “Methods” section: “We performed a systematic review, which was written according to the PRISMA 2020 statement. The study protocol was registered at PROSPERO (CRD42020187354), and there were no subsequent changes to the protocol.”

• The cut-offs used for heterogenity assessment and the choice to quantitatively synthesise are interesting- it would be useful to reference and give an explaination for why this was felt to be warranted as it could be a point of criticism.

o In order to clarify this, we have added the following in the “Synthesis methods” sections of the Methods subheading: “heterogeneity was arbitrarily categorized using cutoff points agreed by the authors as not important (I2 < 40%), moderately significant (I2 = 40%–75%), and considerable (I2 > 75%)”

Review Comments to the Author

Reviewer #1: 

• Gonzales-Huaman and co-authors performed a systematic review about peri-abortion contraceptive counseling interventions.

The introduction builds a logical context for the article, the purpose of which is clear and well defined, appropriate physiopathological background is given, and key concepts are defined. I think the motivation for the present research is clearly expressed.

Data comprised in the tables is clear and consistent with the article’s content. The statistical methods used are appropriate.

The literature is analyzed and critically appraised. Ideas are acknowledged appropriately and accurately; there are no instances of plagiarism, and reference citations are complete and accurate.

o Thank you very much for your kind commentaries.

Reviewer #2:

• 1. The article has not been published elsewhere.

• 2. The statistical analysis has been performed appropriately without performing a meta-analysis for counselling provided by physicians and midwives.

• 3. The article is not presented in an intelligible fashion or is written in standard English.

• 4. It is not necessary

• 5. The article adheres to appropriate reporting guidelines and community standards for data availability

o Thank you very much for your kind commentaries.

Reviewer #3: 

• This is a relevant systematic review which is intended to provide robust evidence regarding the effectiveness of peri-abortion contraceptive counseling in several settings towards reduction of the incidences of repeat abortions.

• The manuscript has been well written with full considerations for the requirements for the condcut and reporting of sytematic reviews and meta analyisis.

• There are however few gramatical errors that needs to be corrected and suggestiosn towards improving the quality of the report.

o We agree. We have performed an in-deep reading of the manuscript and corrected all the typos.

ABSTRACT

• Comment 1: The time frame for the studies included in the systematic review is conspicuously missing. This should be reflected not in the abstract.

o We agree. We added the publication period of the studies in the results section of abstract: “Eleven RCTs were eligible for inclusion (published from 2004 to 2017)”.

• Comment 2: Also include the databases searched by the authors ad whether there were language restrictions in the search.

o We agree. We added the databases to the abstract: “The literature search was performed in June 2021 in PubMed, Central Cochrane Library (CENTRAL), Scopus, and Google Scholar; without restrictions in language or publication date.”

CONCLUSION

• Comment 3: The authors stated in the conclusion “There is no evidence that involvement of different professionals influences contraceptive use”. It is not clear how this was arrived at since only two studies out of 11 compared influence of the involvement of different professionals in counselling. Rather due to the paucity of RCT that compared the two approaches, there is limited evidence that involvement of different professionals influences contraceptive use.

o We agree. We correct the wording of this section. “Also, the studies did not find a difference in the effects of counseling interventions given by different providers.”

Key messages

• Comment 4: The key messages from this study has not been well highlighted in this section. The following statements should be deleted from this section during revision:

o We conducted a systematic review and meta-analysis of 11 RCTs to determine the effect of peri-abortion contraceptive counseling.

o We evaluated the effect of contraceptive counseling on peri-abortion contraceptive use to assess the contribution of this strategy to reducing future pregnancies.

o Replace the above with information that highlights key messages like whether this was the first RCT on the subject matter, uniqueness of study, findings, conclusions, and recommendations:

We agree, and rewritten this section, as follows: 

• “Previous systematic reviews (Ferreira 2009 and Stewart 2015) were performed several years ago and did not assess the certainty of the evidence. Thus, we updated the evidence regarding peri-abortion contraceptive counseling.

• Peri-abortion contraceptive counseling may increase effective contraceptive use, while we did not find significant improvements in other outcomes. Also, we found no evidence of the difference in the effects when comparing contraceptive counseling provided by different providers.

• To date, this is the most comprehensive systematic review that has assessed the effects of peri-abortion contraceptive counseling, summarizing information that may be useful to informed decision-making. Since evidence certainty was very low, future well-designed RCTs are needed.”

.

Introduction

• Comment 5: This statement may not be relevant to the context. Please delete: Page 4, lines 2-3: Globally, abortion is a complex situation for people with uterus (cis, trans and non-binary people). [2].

o We agree. We removed this sentence.

• Comment 6: Page 4, Lines 14-17: Therefore, we performed a systematic review to summarize the available data on randomized controlled trials (RCTs) effects of peri-abortion contraceptive counseling intervention on contraceptive use and obstetric outcomes. 

Paraphrase to: Therefore, we performed a systematic review to summarize the available data from randomized controlled trials (RCTs) that evaluated the effects of peri-abortion contraceptive counseling intervention on contraceptive use and obstetric outcomes

o We agree. We followed your suggestion and added this sentence to the end of the introduction: “Therefore, we performed a systematic review to summarize the available data from randomized controlled trials (RCTs) that evaluated the effects of peri-abortion contraceptive counseling interventions”

Methods

• Comment 7: 

Say: Page 4, line 24, Page 5 lines 1-2: and enhanced the new strategy considered for the trial) and were published as original papers in scientific journals

Should say: and enhanced counselling as the new strategy considered for the trial) and were published as original papers in scientific journals.

o We agree. We edited this sentence as follows: “We defined standard counseling as the intervention regularly practiced in the study context and enhanced counseling as the new intensified strategy considered for the trial.”

Information sources and search strategy

• Page 5, lines 6-8: We performed a literature search of four sources: PubMed, Central Cochrane Library (CENTRAL), Scopus, and Google Scholar (for this, we only assessed the first 100 results). 

Comment 8: It is not clear why only the first 100 results from google scholar was assessed. This should be elucidated upon.

o We clarified this section in the manuscript adding the following in the first paragraph of the "Information sources and search strategy" section:

“Since Google Scholar sorts its results starting with those that have the best match with the search terms, we consider that evaluating the first 100 results would include all relevant studies on the subject in this repository, a methodology that has been used in previous systematic reviews (8-11).”

Outcomes of interest

• Comment 9: The authors have listed various definition of terms considered for this study but has not referenced these considerations for objectivity in assessment of these studies in relation to widely accepted definitions. This should be included in the revision.

o For the purpose of this study, since there are no uniform definitions in the literature, the authors considered the definitions and cut-off points described in the manuscript. In order to clarify this, we have added the following in the second paragraph of the “Outcomes of interest " section: “We considered effective contraceptive methods as any of the following: oral contraceptives, patch, ring, monthly injectable, quarterly injectable, condom, implant, vaginal ring, contraceptive patch, intrauterine device (IUD), intrauterine system (IUS), and sterilization or vasectomy. We considered long-acting reversible contraceptives (LARC) as any of the following: IUD, IUS/hormonal IUD, and implant. We considered obstetric events as either an unintended pregnancy or another induced abortion after counseling.”

Results

• Page 11, lines 1-2: In addition, the proportion of women leaving the facility with at least one contraceptive method was similar in the “nurse group” than in the “physician group”

Comment 10: Change to: In addition, the proportion of women leaving the facility with at least one contraceptive method was similar in both groups

o We agree. We followed your suggestion and edited this sentence.

“did not find significant differences in the proportion of women prescribed contraceptives (nurse group: 99.1%, physician group: 98.7%), in the type of contraceptive prescribed, and in the proportion of women leaving the facility with at least one contraceptive method (96.7% vs 97.3%).”

Discussion

• This section has been well written and exhaustive

• Page 11, line 18: which include six RCTs

Comment 11: which included six RCTs

o We agree. We made this change.

Limitations

• The authors have highlighted the main limitations to this study and interpretation of results.

• However, to date, this is the most comprehensive systematic review that has assessed the effects of peri-abortion contraceptive counseling, summarizing information that may be useful to informed decision-making.

Comment 12: This message should be adapted as part of the key message from this study.

o We agree. We added the following to the key messages: “To date, this is the most comprehensive systematic review that has assessed the effects of peri-abortion contraceptive counseling, summarizing information that may be useful to informed decision-making. Since evidence certainty was very low, future well-designed RCTs are needed.”

References

• The journal names should be written as they are abbreviated in the list of index medicus for all the references quoted.

o We agree. We edited the references as suggested.

Reviewer #4:

INTRODUCTION

• There are some grammatical errors that need to be corrected. The aim of the study in the last paragraph of the introduction on page 4 is not clear.

o We agree. We removed this sentence as it was confusing.

RESULTS: 

• Wrong use of tenses and few grammatical errors.

o We agree. We proofread this section with the help of a native English language writer.

DISCUSSION: 

• In page 12, paragraph 3, in the 1st sentence, what do the authors mean by ' the study had more weight'?

o We agree. It was clarified by putting the weight in the forest plot (45.4%).

Reviewer #5:

Methods:

• In the first paragraph under the under the sub-heading ‘outcome of interest’-page 6, authors should kindly note that all injectable contraceptives are hormonal. So the monthly and quarterly injectables are implied. Authors should kindly take note of this.

• Again, Intrauterine system (IUS) is used interchangeable with hormonal IUD.

o Thank you very much for both commentaries, we agree and therefore we have edited the text according to your comment. We eliminated some redundant contraceptives:

“We considered effective contraceptive methods as any of the following: oral contraceptives, patch, ring, monthly injectable, quarterly injectable, condom, implant, vaginal ring, contraceptive patch, intrauterine device (IUD), intrauterine system (IUS), and sterilization or vasectomy. We considered long-acting reversible contraceptives (LARC) as any of the following: IUD, IUS/hormonal IUD, and implant”.

• Language editing is recommended in the 2nd paragraph under the sub-heading ‘outcome of interest’. Specifically “we considered standard care or routine counseling to all interactions with patients preestablished by the hospital, institution or the research group”

o We agree. We edited the text according to your comment.

Limitations and Strengths:

• Were there any limitations in the area of language bias and how was it addressed since the various studies were undertaken in different continents with heterogeneous cultural diversities.

o Although we had no language restrictions, at the end all included RCTs were published in english. To clarify this, we have added the following sentence in the second paragraph of the “study characteristics” subheading: “All were published in full-text in english.”

Additional Comments

• Inclusion of line numbers on manuscript could have been helpful for the review processes.

o We agree. We made the changes according to the PlosOne templates.

Reviewer #6: 

• manuscript needs to be shortened. Authors should also engage a statistician. Consider using the forest plot. also can you include a table that shows the articles reviewed and results at a glance? thank you for great review

o We agree. We edited the text based on the previous comments and tried to reduce the number of words as possible. We have included the forest plots as the figure 2.

---

## [Editor Report · Decision Letter 1]

17 Nov 2021

Peri-abortion contraceptive counseling: a systematic review of randomized controlled trials

PONE-D-21-29042R1

Dear Dr. Taype-Rondan,

We’re pleased to inform you that your manuscript has been judged scientifically suitable for publication and will be formally accepted for publication once it meets all outstanding technical requirements.

Kind regards,

Dylan A Mordaunt, MB ChB, FRACP, FAIDH

Academic Editor

PLOS ONE

Additional Editor Comments (optional):

Thank you for your resubmission and for addressing all of the suggestions. This meets the criteria for publication.
---

## [Editor Report · Acceptance letter]

14 Dec 2021

PONE-D-21-29042R1 

Peri-abortion contraceptive counseling: a systematic review of randomized controlled trials 

Dear Dr. Taype-Rondan:

I'm pleased to inform you that your manuscript has been deemed suitable for publication in PLOS ONE. Congratulations! Your manuscript is now with our production department. 

Kind regards, 

on behalf of

Dr. Dylan A Mordaunt 

Academic Editor

PLOS ONE